# Innate Resistance and Phosphite Treatment Affect Both the Pathogen’s and Host’s Transcriptomes in the Tanoak-*Phytophthora ramorum* Pathosystem

**DOI:** 10.3390/jof7030198

**Published:** 2021-03-09

**Authors:** Takao Kasuga, Katherine J. Hayden, Catherine A. Eyre, Peter J. P. Croucher, Shannon Schechter, Jessica W. Wright, Matteo Garbelotto

**Affiliations:** 1Crops Pathology and Genetics Research Unit, Agricultural Research Service, Davis, United States Department of Agriculture, CA 95616, USA; tkasuga@ucdavis.edu; 2Department of Environmental Science, Policy, & Management, University of California, Berkeley, CA 94720, USA; KHayden@rbge.org.uk (K.J.H.); catherine.eyre@gmail.com (C.A.E.); pete@petercroucher.com (P.J.P.C.); sapphiresps@gmail.com (S.S.); 3Royal Botanic Garden, Edinburgh EH3 5NZ, UK; 4United States Department of Agriculture, Forest Service, Pacific Southwest Research Station, Davis, CA 95618, USA; jessica.w.wright@usda.gov

**Keywords:** phosphonate, *Notholithocarpus densiflorus*, Sudden Oak Death (SOD), in planta RNA-Seq, plant defense, gene set enrichment analysis (GSEA)

## Abstract

Phosphites have been used to control Sudden Oak Death; however, their precise mode of action is not fully understood. To study the mechanism of action of phosphites, we conducted an inoculation experiment on two open-pollinated tanoak families, previously found to be partially resistant. Stems of treatment group individuals were sprayed with phosphite, and seven days later, distal leaves were inoculated with the Sudden Oak Death pathogen *Phytophthora ramorum*. Leaves from treated and untreated control plants were harvested before and seven days after inoculation, and transcriptomes of both host and pathogen were analyzed. We found that tanoak families differed in the presence of innate resistance (resistance displayed by untreated tanoak) and in the response to phosphite treatment. A set of expressed genes associated with innate resistance was found to overlap with an expressed gene set for phosphite-induced resistance. This observation may indicate that phosphite treatment increases the resistance of susceptible host plants. In addition, genes of the pathogen involved in detoxification were upregulated in phosphite-treated plants compared to phosphite-untreated plants. In summary, our RNA-Seq analysis supports a two-fold mode of action of phosphites, including a direct toxic effect on *P. ramorum* and an indirect enhancement of resistance in the tanoak host.

## 1. Introduction

Phosphites, salts or esters of phosphonic acid, are systemic compounds first shown to be highly effective against diseases caused by oomycetes in the 1970s (reviewed in [1]) and have since been used widely as fungicides in horticulture and natural ecosystems [2,3]. Nonetheless, the precise mode of action of these compounds is not fully understood. It has been postulated that phosphite compounds may have both direct and indirect effects on the oomycete *Phytophthora* spp. At high doses, they act directly on pathogen growth and sporulation [4,5,6], while at low doses they stimulate host defenses, including an increase in plant defensive enzymes [4,5,7,8,9]. Phosphite’s stimulation of extant host defenses has been hypothesized to be the reason for the compounds’ varying actions among individuals [7,10,11].

First recognized in the mid 1990s [12,13], the invasive pathogen *Phytophthora ramorum* has rapidly spread among coastal forests of California and Oregon, killing tens of millions of trees. Tanoaks are among the most susceptible wild North American hosts to the pathogen, and experience both the deadly but rarely infectious bole form of the disease this pathogen causes (the widely known “Sudden Oak Death”), and the foliar and infectious form of the disease known as “Ramorum Blight” [14]. Tanoak has experienced the highest mortality rates among wild hosts [15,16,17], to the point that it is widely speculated that extinction of local populations may soon occur in heavily infested areas [18,19]. Agri-fos (Agrichem, potassium phosphite salts) is a registered fungicide in California and is now used for the protection of trees from Sudden Oak Death (causal organism the oomycete *Phytophthora ramorum*) in wildlands and particularly for the protection of oaks (*Quercus* spp.) and tanoaks (Fagaceae: *Notholithocarpus densiflorus* (Hook. and Arn.) Manos, Cannon and S. Oh) [13,20,21].

While the ecological and cultural importance of tanoaks is well established [22], owing primarily to their low commercial value, tanoaks have not been widely propagated for forestry. There are no extant plantations of tanoaks with a known multigenerational pedigree, limiting the scope of study of disease resistance traits using traditional quantitative genetic techniques. For example, the development of quantitative trait loci from phenotypic observation and known family relationships is impossible. Since the onset of the Sudden Oak Death epidemic in California and Oregon, the potential utility of known breeding stocks for the study of the genetics of pathogen resistance and other phenotypic traits has been recognized, and a USDA-Forest Service-Pacific Southwest Research Station-funded common garden study of open-pollinated seed has been established in Berkeley, CA [20,23]. The common garden research has resulted in the identification of maternal family groups with signs of disease resistance, including some with markedly lower infection rates. Note that the use of “resistance” and “tolerance” is nonstandard in plant pathology. In this paper, resistance refers to the presence of host mechanisms that either limit the establishment of infection or limit the spread of an infection within a host, sensu Roy and Kirchner [24]. Combining these two phenotypes, i.e., reduced disease incidence and reduced disease severity, in the same category of “disease resistance” makes sense for the tanoak-*P. ramorum* pathosystem, given our previous results showing that tanoak families displaying reduced disease severity in the laboratory had a higher rate of survival in field trials, due to reduced disease incidence [20]. Despite the absence of commercial forestry sources of germplasm, there are considerable genomic resources available in this pathosystem. Notably, there are both a *P. ramorum* genome sequence [25], and a de novo assembled transcriptome (a reconstructed transcriptome from RNA-Seq experiments) of tanoak [26].

Another complication of the tanoak-*P. ramorum* pathosystem is the lack of established clonal propagation, preventing replication using identical genotypes, combined with the presence of quarantine regulations that require the destruction of any inoculated plants. Thus, it is currently impossible to know whether individual trees will demonstrate resistant or susceptible reactions prior to inoculation. To overcome this barrier, we used previously identified open-pollinated family groups which carried resistance, that is in which approximately 20% of offspring demonstrated a resistant phenotype (dieback 25% or less) [20]. Hereafter, the resistance phenotype of phosphite-untreated tanoak is defined as “innate resistance” to distinguish it from phosphite-induced resistance. 

Here we examine gene expression in seedlings exposed to *P. ramorum*, both with and without the application of phosphonate. We hypothesized that phenotypic differences on the outcome of infection correlate with differences in regulation of key genes before as well as after the infection takes place. Hence, comparisons among transcriptomes may lend insights into the genes involved in innate tanoak resistance and into the mode of action of phosphites. In addition, in plant pathology research, disease phenotypes are monitored in detail; however, actual biomass of pathogens, their physiological states in planta, and gene regulation have been less studied due to the difficulty of in planta observation. Taking advantage of deep sequencing technology, we have also investigated gene activity of the pathogen in inoculated tanoaks. We hypothesized that phosphite in the plant tissue perturbs the transcriptome of *P. ramorum* in planta and changes in gene activity would help to elucidate the mode of action of phosphites. 

## 2. Materials and Methods 

### 2.1. Experiment Overview

We chose two open-pollinated tanoak half-sibling family groups (Family 10 and 12) that carried innate resistance, i.e., families that were previously tested and had shown innate resistance to *P. ramorum* [20]. Multiple inoculations were performed on previously unchallenged members of the two families, with leaves harvested and flash-frozen for RNA extraction before and one week after inoculation. The remaining inoculated leaves were left intact, and the trees were followed over the course of five weeks to determine disease phenotype. The design and the phenotypic results allowed us to study gene expression during disease response in phosphite-treated resistant hosts (in which the treatment worked as expected), in phosphite-treated but susceptible hosts (in which phosphite was not effective nor was there innate resistance), and in untreated naturally susceptible or resistant trees.

### 2.2. Experimental Design for Gene Expression Analysis

Tanoaks for inoculation were selected from a container garden located in Berkeley, California, housed outdoors under 50% shade. Seedlings had been grown from seed in 6.9 cm × 25.4 cm cones, and at one year of age potted up to 10.2 cm × 10.2 cm × 30.5 cm pots, all in UC mix growing medium [27]. A total of 272 seedlings from 34 open pollinated families in three California regions were prescreened using inoculation experiments. Two families (10 and 12) were selected on the grounds of prior observation of disease resistance levels up to 25% per family, based on the intact stem inoculation method described in [20]. For each open-pollinated sampling family, 4-year old seedlings were randomly assigned to treatments (P, phosphite) or control (C, no phosphite) groups (Figure 1). Within each treatment group, individuals were randomly assigned to be inoculated with a *P. ramorum* zoospore suspension or to the non-inoculated control. There was one non-inoculated control per family and treatment. 

### 2.3. Phosphite Treatment

We have previously shown that bark application of phosphites was ineffective whereas bark applications of phosphites with the organosilicate surfactant Pentra-bark™ (Agrichem, Medina, OH, USA) were consistently effective in suppressing colonization by *P. ramorum* without causing observable phytotoxicity [28]. Plants were treated with a 2.4 M potassium phosphite solution (48.75% *v*/*v* Agri-Fos, Agrichem, Medina, OH, USA) mixed with a 2.5% surfactant (Pentra-Bark, Agrichem Manufacturing Industries) seven days prior to inoculation. Hereafter, “phosphite treatment” refers to the treatment with phosphite and the surfactant solution. The treatment was applied by hand ensuring coverage of 10 cm of each stem upwards from the soil line, with care taken to avoid application directly to leaves. Control plants were sprayed with deionized H_2_O.

### 2.4. Inoculum and Leaf Harvest

*P. ramorum* isolate MK1461, first isolated from a California bay in San Mateo County, California, belonging to the NA1 lineage, was found previously to be of intermediate but consistent aggressiveness, and was used for all tanoak inoculations. To prepare zoospores for inoculation, cultures were grown on 10% *v*/*v* clarified V8 agar [29] for 14 days at 18 °C. To induce sporangial formation, cultures were cut into squares approximately 1 cm^2^, and placed into empty petri plates. Plates were flooded with a sterile 1% *w*/*v* soil extract solution (soil tea) and incubated in the dark at 18 °C for 48 h, until sporangia were formed. Zoospore release from sporangia was induced by cold shocking the cultures as follows. The mycelial squares incubated in multiple petri plates were consolidated into a single vessel before being placed in ice water for 30 min. After that, the mycelial squares were further incubated at room temperature for one hour. Zoospores were quantified using a hemocytometer, and diluted with sterile soil tea to a final concentration of 5 × 10^4^ zoospores per mL. This zoospore suspension was used immediately for seedling inoculations. 

After the phosphite treatments, all seedlings were transferred to growth chambers with a 16-h light (6 a.m. to 10 p.m.): 8-h dark cycle at 18 °C, and watered every 2 days with supplemental regular misting to maintain humidity. At seven days after phosphite treatment, two leaves were removed and flash-frozen in liquid nitrogen (T0) and stored at −80 °C. Each seedling was then lightly wounded with a scalpel and inoculated at leaf axils. Then, 100 µL of zoospore suspension was applied to the leaf axil and held in place by a small wax cup (Parafilm M, Bemis Company, Neenah, WI, USA) formed at the base. One seedling per each family and treatment was randomly selected for a mock inoculation with sterile soil tea as a negative control. The number of leaf axils was determined by seedling architecture: where 6 distinct leaves (not part of the same whorl) capable of holding inoculum in this way were identified, 5 were inoculated. One non-inoculated control had only 4 “inoculation” points. At seven days post-inoculation (T7), 4 inoculated leaves per tree were collected, flash-frozen in liquid nitrogen, and stored at −80 °C. In order to minimize diurnal variation of the transcriptome, all leaves were harvested between 1 pm and 4 pm. Trees were maintained under the same conditions for five weeks post-inoculation. After five weeks (T35), seedlings were assessed for dieback using a scale 0–4 based on quartiles of percentage of dead above-ground tissue: 0 = no dieback, 1 = 1–25%, 2 = 26–50%, 3 = 51–75%, 4 = 76–100%. 

### 2.5. RNA Extraction

Leaves were subsampled while frozen by excising small sections from the inoculation point around the midrib at the base of each leaf, approximately encompassing the area exposed to inoculum. Samples were ground to powder while frozen in Lysing Matrix A tubes (MP Biomedicals, Irvine, CA, USA), with an additional ceramic bead in each tube, using an MP Biosystems FastPrep with CryoPrep^TM^ attachment. RNA was extracted using a CTAB-Chloroform-Isoamyl alcohol extraction and lithium chloride precipitation [30,31]. Following precipitation, pellets were air dried and subjected to a further clean-up using a ZR RNA MiniPrep^TM^ (Zymo Research, Irvine, CA, USA), including an on-column DNase treatment. Total RNA integrity was assessed using an Agilent 2100 Bioanalyzer (Agilent Technologies, Santa Clara, CA, USA). Although tanoak samples were pulverized on dry ice, which was followed by a CTAB extraction protocol, RNA Integrity Numbers (RINs) evaluated by Agilent bioanalyzer were often low (median 4.7). Unlike mammalian samples, plant cells have chloroplast ribosomes and a variable ribosomal RNA size, which can lower the RNA RIN value even when purified RNA is intact [32]. In addition, samples with necrotic tissues were expected to contain some degraded RNA from the host as well as the pathogen. We hence proceed to RNA-Seq cDNA library construction regardless of the RIN value and RNA-Seq was carried out. Integrity of RNA was then bioinformatically evaluated and samples with degraded RNA were removed from the dataset (see the next section). 

### 2.6. Tanoak RNA-Seq cDNA Library Construction

A total of 46 plant samples were used for library construction. Sufficiently high quantity of RNA (yield above 0.1 μg) was extracted from 44 of 46 samples of the combined tanoak and *P. ramorum* tissue, as described above and was used to prepare RNA-Seq libraries using Illumina TruSeq v2 (Illumina, San Diego, CA, USA). Twelve samples per lane were indexed, multiplexed, and sequenced on a HiSeq2000 as 100 bp paired-end runs at QB3 Genomics Sequencing Laboratory at UC Berkeley. Transcript Integrity Numbers (TINs) were then calculated using RSeQC package version 2.6.4 [33] after RNA-Seq reads were aligned to the tanoak de novo transcriptome assembly [26] with align function in the Rsubread package [34], which was run on R 3.3.3 statistical software [35]. TIN values range from 0 (the most degraded) to 100 (the most intact). As median TIN scores of most samples were high (median 80.3), we judged that mRNA integrity of most transcripts were sufficiently high for reliable RNA-Seq analysis (Appendix A). 

### 2.7. Bioinformatics Pipeline

#### 2.7.1. Tanoak Expression

By first aligning sequences to the *P. ramorum* transcriptome, we were able to subtract both *Phytophthora* and highly conserved sequences, leaving only tanoak transcripts. Briefly, the “view” function in SAMtools version 1.9 [36] with options -u -f 12 -F 256 was used to subtract *P. ramorum* transcripts from the RNA sequence files mapped to the *P. ramorum* assembly in the BAM format. Resulting BAM files were subsequently converted to fastq file format using the “bam2fq” function in SAMtools. 

The genome sequence of tanoak is not yet available and the gene set represented in the de novo transcriptome of tanoak [26] is likely incomplete. The genome of English oak *Quercus robur*, the most closely related species whose genome has been sequenced, was used as a reference genome. Gene Ontology (GO) terms [37,38] and Kyoto Encyclopedia of Genes and Genomes (KEGG) pathways [39], implemented in Blast2GO [40] were used to annotate the English oak genome (Appendix A). The tanoak-only RNA sequences were aligned to the English oak assembly Qrob_PM1N.fa.gz (https://urgi.versailles.inra.fr/download/oak/Qrob_PM1N.fa.gz, accessed on 8 March 2021) [41] with the align function in the Rsubread package [34]. The mapped reads were then counted using the featureCounts function in the Rsubread package with a gene coordinate file in SAF format derived from an English oak gene model coordinate file, Qrob_PM1N_genes_20161004.gff. The Pearson’s correlation coefficient between global mRNA expression patterns was used to cluster tanoak transcriptomes using the hclust function [42] with the average linkage option in R 3.3.3 statistical software. DESeq2 [43] was used for expression quantification between transcriptome clusters and treatments. The default values of the parameters and workflows outlined in software documentation were used. A total of four comparisons were analyzed for differential expression. 

#### 2.7.2. *P. ramorum* Expression

RNA sequences were aligned to the *P. ramorum* assembly ramorum1.allmasked (https://genome.jgi.doe.gov/portal/Phyra1_1/Phyra1_1.download.ftp.html, accessed on 8 March 2021) [25] with align function in the Rsubread package [34]. The mapped reads were then counted across the *P. ramorum* genes using the featureCounts function in the Rsubread package with a gene coordinate file in SAF format derived from a *P. ramorum* gene model coordinate file, FM_Phyra1_1.gtf. Uninfected tanoak transcriptomes were also mapped to the *P. ramorum* genome and sixteen *P. ramorum* genes to which tanoak transcripts were mapped were removed from the dataset. The remaining transcripts were used for further analysis. Differentially expressed genes between groups were estimated using DESeq2 [43] with the un-normalized gene count dataset from featureCounts. A false discovery rate cut off <0.05 was used to filter differentially expressed genes. *P. ramorum* gene models (ramorum1.proteins.fasta in https://genome.jgi.doe.gov/portal/Phyra1_1/Phyra1_1.download.ftp.html, accessed on 8 March 2021) were annotated with gene ontology (GO) terms [37,38] and KEGG pathways [39] as implemented in Blast2GO [40].

For the *P. ramorum* and tanoak datasets, a gene set enrichment analysis (GSEA) [44] was used to evaluate over- or under-representation of functional categories (GO, or KEGG pathway) across expression gene clusters or differentially expressed gene sets using Fisher’s exact test function fisher.test() with the statistical software R 3.3.3 or a same function implemented in Blast2GO. The false discovery rate according to Benjamini and Hochberg [45] was used for multiple hypothesis correction (adjusted *p*-value < 0.05). 

#### 2.7.3. Availability of Data and Material

BAM files for the original Illumina RNA sequencing data aligned to the de novo tanoak transcriptome library and the same data aligned to *P. ramorum* reference genome were deposited in the NCBI Sequence Read Archive under study accessions SRP157197 and SRP157863, respectively.

## 3. Results

### 3.1. Tanoak Families Differed in the Presence of Innate Resistance and in the Effectiveness of Phosphite Treatment

Two maternal families of tanoak, previously characterized as having 12.5–25% of their offspring resistant to zoospore inoculation by *P. ramorum* isolate MK1461, were examined for disease resistance and phosphite response (Figure 1) (see Methods for details). Leaves of phosphite-treated and control seedlings were sampled prior to inoculation (T0 samples) and seven days post inoculation (T7 samples). Our previous work has showed that efficacy of the systemic fungicide is consistently seen seven days post treatment [46]. At T7, foliar lesions were seen in many inoculated leaves, whereas foliage dieback symptoms were not yet seen. Tanoak seedlings challenged with *P. ramorum* revealed a range of responsiveness to phosphite treatment (Figure 2). Family 10 was the most resistant in the absence of phosphite, with 5 out of 9 seedlings having mild or zero dieback following inoculation. Notably, Family 12 had a greater frequency of dieback (6/8) without treatment, but 0 of 8 phosphite-treated seedlings showed severe dieback symptoms. 

A total of 44 RNA samples were sequenced and transcript integrity number (TIN) was estimated for each sample to measure the level of RNA integrity (Supplementary S1). It was found that median TIN scores, which correlate well with RNA integrity number (RIN), were sufficiently high (above 75) for 34 of the 44 samples. All but one sample showing TIN scores below 75 were removed from the dataset. The only sample having a low TIN score (49.5%) was a T7 sample, which showed the highest percentage of *P. ramorum* reads (13.9%) and was used only for hierarchical clustering and in planta transcriptome analysis of the pathogen. 

### 3.2. Clustering of Expression Revealed Factors Influencing Tanoak Transcriptomes

Of the 35 transcriptomes, 15 were from samples at T0 (seven days after phosphite treatment, just before inoculation), and 20 were from samples at T7 (seven days after inoculation). After the subtraction of *P. ramorum* transcripts, the 35 tanoak transcriptomes were mapped to the English oak genome and were hierarchically clustered according to their global expression patterns (Figure 3 and Figure 4). 

Hierarchical clustering analysis showed three distinctive clusters. Cluster A and cluster B contained only samples collected at T7 and never included uninoculated controls (marked with “N”). Cluster A contained primarily T7 samples with susceptible phenotypes (Fisher’s exact test *p* = 0.031, Appendix A). In comparison to Cluster A and B, Cluster C contained relatively homogenous transcriptomes comprising all the T0 samples (resistant or susceptible, Figure 4a) and a part of T7 samples with a predominance of resistant phenotypes (dieback 0–25%). Transcriptomes of all non-inoculated controls at T0 and T7 were also found in Cluster C. Cluster C contained primarily uninfected samples (all T0 samples and non-inoculated T7 samples, Fisher’s exact test, *p* < 0.001, Appendix A). In summary, the hierarchical clustering of tanoak transcriptomes was associated with disease symptoms and infection status. The effect of phosphite treatment was not readily recognizable. 

Of the total of 25,808 predicted genes in the English oak genome, 14,964 genes (58%) found highly similar sequences in the tanoak de novo transcriptome assembly at DNA sequence identity of 70% or larger. Differentially expressed genes (DEGs) between clusters were then analyzed in pairwise comparisons. Between clusters (B and C) and Clusters (A and C), 11,102 and 13,641 DEGs were identified respectively, of which 8460 genes were overlapping with the same directions of fold changes. Because a large portion overlaps, transcriptomes in Cluster A and B were combined, and DEGs between the combined transcriptomes (cluster AB) and cluster C were analyzed. As a result, 13,443 DEGs were identified, of which 11,502 genes (86%) overlapped with DEGs in the clusters A and C comparison. 7178 and 6265 genes were upregulated and downregulated in the cluster AB (Table 1, Appendix A). 

A gene set enrichment analysis (GSEA) was then employed to elucidate physiological differences between clusters AB and C observed in transcripts mapped to the English oak genome. GSEA on KEGG pathways indicated upregulation of genes involved in phenylpropanoid biosynthesis and phenylalanine metabolism, which include two genes encoding for phenylalanine ammonia-lyases (PAL), enzymes involved in the first step of the biosynthesis of several phytoalexins and lignin [47]. On the other hand, genes involved in photosynthesis, starch and sucrose biosynthesis were downregulated in cluster AB (i.e., upregulated in cluster C, Table 2). GSEA on GO terms was partly overlapping GSEA on KEGG pathways, highlighting upregulation of genes involved in defense (high in AB) and downregulation of those in photosynthesis. It is noteworthy that defense-related GO terms such as jasmonic acid metabolic process, and chitin catabolic process were enriched in cluster AB. In addition, the GO enrichment analysis indicated upregulation of genes for respiration and energy generation in cluster AB. Downregulation of photosynthesis genes and upregulation of genes for energy generation are hallmarks of plant immune processes [48,49,50]. In conclusion, GSEA showed that transcriptomes in cluster AB represent infection and plant defense whereas those in cluster C represent free of infection or disease with minor symptoms. The DEGs and results of GSEA are shown in Appendix A and Table 2, respectively.

### 3.3. Search for Tanoak Transcriptome Signatures Associated with Innate Resistance and Phosphite Treatment

Our cluster analysis did not immediately identify innate resistance or phosphite induced transcriptome patterns, however, transcriptome patterns of T7 samples seemed to associate with the occurrence of *P. ramorum* transcripts. Among the eighteen inoculated T7 samples, seven tanoak leaf samples yielded cDNA sequence reads from *P. ramorum* which were above the baseline (Figure 5). The number of *P. ramorum* cDNA sequences mapped to the *P. ramorum* genome at T7 did not correlate with the observed disease responses, which were scored 35 days after inoculation (T35) and 28 days after T7 sample collection (Figure 5, Fisher’s exact test *p* = 0.271, Appendix A). Although phosphite-untreated susceptible plants showed a high level of *P. ramorum* reads, the difference between phosphite-treated and untreated susceptible samples was not statistically significant (Mann Whitney U test *p* = 0.095, Appendix A). Detection of *P. ramorum* transcripts was however associated with hierarchical clusters. All seven samples with detectable levels of *P. ramorum* transcripts were found in either Cluster A or B (Fisher’s exact test *p* = 0.004), whereas samples with *P. ramorum* transcripts below threshold were found in all three clusters. In summary, global mRNA expression pattern of tanoak at T7 was most strongly associated with active growth of the pathogen inside the host tissue, but not with disease phenotypes at T35 (*p* = 0.271) or phosphite treatment (*p* = 0.370) (Figure 4b, Appendix A). 

The other aspect revealed by the hierarchical clustering analysis was that disease phenotypes at T35 did not always correlate with disease progression at T7. For instance, at T7, one out of seven susceptible plants showed global mRNA profiles associated with no infection (Library HS1A_index5 Figure 4b). On the other hand, four out of eleven resistant plants showed disease associated global mRNA profiles at T7 (cluster A or B in Figure 4b), and *P. ramorum* transcripts were detected in three of the four resistant samples. The observed inconsistency between transcriptomes and disease phenotypes indicated a large variation in disease progression among samples at T7, which can confound interpretation of transcriptome analysis. We therefore searched for the signature of innate resistance and phosphite-induced resistance only in the samples before inoculation (T0 samples) (Table 1). Due to small representation, Family 10 seedling samples were excluded from data analysis. 

Untreated susceptible trees (Control susceptible: Cs) and untreated resistant trees (Control resistant: Cr) before inoculation were compared in search of a transcriptome signature for innate resistance (Cs and Cr at T0 in Table 1, Appendix A). The comparison using the English oak genome as a reference yielded 466 DEGs, of which 268 were upregulated in Cr plants. Interestingly, GSEA implicates enrichment of defense-related genes such as “Phenylpropanoid biosynthesis”, “terpenoid biosynthetic process”, “Biosynthesis of antibiotics”, and “oxylipin biosynthetic process” are enriched in genes upregulated in resistant plants (Table 3). 

Effects of phosphite on transcriptomes were examined through a comparison of phosphite treated (seven days post treatment) and phosphite untreated samples at T0 (P and C plants at T0). No differentially expressed genes were detected (Table 1, Appendix A). 

Although its mode of action is not well understood, phosphites have been used to protect susceptible plants from pathogens. In order to highlight which genes may be involved in phosphite-induced resistance, susceptible untreated trees were compared to resistant phosphite-treated trees (Cs and Pr at T0 in Table 1, Appendix A). Note that owing to difficulty in clonal propagation of tanoak, innate resistance of Pr plants were not evaluated. However, judging from the low frequency of innate resistance, most of Pr trees from Family 12 were unlikely to be innate-resistant to *P. ramorum*. Family 12 trees were the highest responders to the phosphite treatment: six out of eight untreated trees were susceptible, whereas all eight phosphite-treated trees were resistant to *P. ramorum* (Figure 2). Thirty-one genes were upregulated and 16 genes were downregulated in phosphite-treated resistant (Pr) trees (Table 1). 

It was found that over half of DEGs (22 out of 31 DEGs) in the Cs Pr comparison are also DEGs in the Cs Cr comparison (Table 4). In other words, the changes in gene expression patterns observed in Cs plants following phosphite treatment (i.e., DEGs between Cs and Pr) are positively correlated with the difference in gene expression between Cs and Cr plants. Enrichment of one GO term associated with phloem development as well as the triterpenoid biosynthetic process were found to be overrepresented among DEGs shared between Cr and Pr in comparison to Cs in Family 12 (Table 3) and two DEGs predicted as “SIEVE ELEMENT OCCLUSION B-like” were annotated with the GO term phloem development. Three genes involved in flavonoid modifications and four genes encoding LRR receptor-like serine threonine kinases (RLKs) were also upregulated in Cr as well as Pr plants (Appendix A). These proteins have been implicated in active defense. 

### 3.4. Analysis of in Planta Phytophthora Ramorum Transcriptomes

Although the percentage of transcripts from the pathogen was low, Illumina RNA-Seq yielded between 96,647 to 2,713,332 reads for seven T7 samples. We tested for DEGs in in planta *P. ramorum* transcriptomes between phosphite-treated and untreated samples. Consequently, 20 DEGs, all upregulated in phosphite-treated samples, were identified (Appendix A). GO enrichment analysis identified “pyridoxal phosphate (Vitamin B_6_) biosynthesis process” among phosphite upregulated genes (false discovery rate corrected *p* = 0.015). Two genes for Vitamin B_6_ biosynthesis are known for involvement in detoxification. In addition, four ATP-binding cassette (ABC) transporters and one major facilitator superfamily (MFS) transporters, groups of genes often involved in drug resistance and detoxification [51], were identified. It is noteworthy that phosphite was applied to stems at the soil line, whereas transcriptomes were derived from inoculated leaves distant from the soil line. We did not quantify phosphite in the inoculated leaves, however, a strong influence of the systemic fungicide on the *P. ramorum* transcriptomes was revealed. 

## 4. Discussion

Dual RNA-Seq in combination with hierarchical clustering and gene set enrichment analysis revealed a complicated interplay of *P. ramorum*, phosphite, and genetically heterogeneous tanoak individuals from a wild population. These analyses indicated a large variation in disease progression at T7 regardless of the genetic background of tanoak or phosphite treatment. In addition to the genetic makeup of host plants and phosphite treatment, age as well as developmental stage and history of microenvironmental conditions can inevitably influence the structural integrity of each leaf, and thus susceptibility to zoospores used as inocula. For instance, at T7, most resistant plants showed global mRNA profiles associated with healthy plants while others showed those associated with infection. This indicates that most resistant plants fended off the pathogen at the early stages of infection, perhaps as early as the time of pathogen entry, and thus either their transcriptomes had not been perturbed or they had come back to their basal state (Cluster C) at the time of sampling. In other cases, the pathogen may have initially invaded the plant tissue causing disease associated changes in expression (Cluster A or B), but the pathogen was then later cleared or contained, and failed to become established in resistant plants by T35. Transcriptomes of phosphite-treated samples at T7 were also diverse, and did not correlate well, regardless of their resistant or susceptible phenotype at T35. Hierarchical clustering analysis, as we demonstrated, is thus particularity useful to evaluate variation and reproducibility of disease-associated transcriptomes of non-clonal and/or developmentally heterogenous samples. 

### 4.1. Use of Organosilicate Surfactant for Phosphite Application

Garbelotto et al. [28] reported that bark applications and soil drenches of unamended phosphites were not effective against Sudden Oak Death, while foliar applications of phosphites amended with a range of surfactants caused excessive phytotoxicity in oaks and tanoaks. Two treatment approaches instead were highly effective; injection of highly concentrated phosphites directly into the stem, a procedure that requires stems of at least 4 cm in diameter, and bark applications of phosphites with Pentra-Bark, a treatment that is feasible on smaller trees as well and thus was chosen as the only possible approach here, given the small size of the plants. Considering that phosphites are only effective when used with Pentra-Bark in this pathosystem, and based on the fact that this organosilicate surfactant is known to strongly adhere on the outer bark of the tree without any presumed effects on the pathogen or on tree physiology [21], we decided not to test the two compounds in separate treatments, given that bark applications of phosphites only or of Pentra-Bark only are not registered in California for the treatment of Sudden Oak Death in oaks and tanoaks. While an effect of Pentra-Bark on pathogen or host is unlikely, it is important to note that the effects measured and reported in this study were caused by the combined application of both compounds. When phosphite-treated (with Pentra-Bark) and phosphite-untreated samples were compared (C and P plants at T0 in Table 1), no deferentially expressed genes were detected. Noise in gene expression due to Pentra-Bark is thus likely low.

### 4.2. Genes Involved in the Innate Resistance

Due to a large variation in disease progression at T7, only T0 (pre-inoculation) samples were used to search for transcriptome signatures associated with innate resistance. Previous work indicates that disease resistance to Ramorum Blight in tanoak is quantitative (quantitative disease resistance, QDR) [20]. QDR is under the control of multiple and diverse classes of genes each with small effects [52,53]. QDR genes downstream to the event of MAMPs perception could be at a basal low expression state before inoculation, therefore they might not be detectable through the transcriptome comparison of uninoculated resistant and susceptible plants. Comparison of Cr and Cs plants at T0 (uninfected and phosphite-untreated) yielded 466 DEGs. Gene set enrichment analysis showed that before inoculation, defense related genes are upregulated in innate resistant plants (Cr). For instance, terpenoids, chorismate, oxylipins, phenylpropanoids and antibiotics (Table 3) all directly or indirectly participate in production of defense metabolites [54,55,56]. Enrichment of GO term “negative regulation of endopeptidase activity” showed the smallest *p*-value (*p* < 0.001). Enrichment of this GO term has also been observed in a disease-tolerant rootstock of avocado plant in response to a fungal root pathogen *Rosellinia necatrix* in comparison to susceptible rootstocks [57] as well as in a disease-resistant spinach cultivar in response to an oomycete downy mildew pathogen *Peronospora effusa* in comparison to a susceptible cultivar [58]. 

### 4.3. Phosphite-Induced Resistance

As for the innate resistance, phosphite-induced resistance in tanoak is likely governed by quantitative trait loci (QTL). When phosphite treated samples and untreated samples at T0 (seven days post treatment) were compared, no DEGs was identified. In the *P. infestans* potato pathosystem, phosphite rapidly induced a transcriptome shift within 3 h, the effect, however, lasted less than 24 h [59]. It is possible that phosphite-induced transcriptome alteration and priming of systemic acquired resistance had taken place before our first sampling at seven days post-treatment. Hence, sampling at earlier time points, as early as three hours might have differentiated variations in phosphite-induced resistance. Comparison of Cs and Pr plants at T0 (seven days post treatment) identified 47 DEGs—of these, 25 were shared with DEGs in the Cs and Cr plant comparison. Of the three phosphite-treated Family 12 trees, which showed the resistant phenotype (Table 1), the number of innate resistant plants is unknown due to unavailability of clonal propagation. If the frequency of innate resistant seedlings in Family 12 is 1/4 as observed, it is highly unlikely that phosphite-treated plants showing resistance are due only to innate resistance (*p* = (1/4)^3^ = 0.015). Shared DEGs between (Cr v. Cs) and (Cs v. Pr) comparisons in Family 12 (Table 4) were likely due to the effect of phosphite on naturally susceptible plants. 

Diverse classes of DEGs are shared between Pr and Cr plants. Upregulation of three genes for sieve element occlusion proteins and two genes for biosynthesis of triterpenoid beta-amyrin are attributable to the enrichment of GO terms “phloem development” and “triterpenoid biosynthetic process”, respectively (Table 3). Homologs of sieve element occlusion proteins have shown to limit phloem mass flow in response to pathogen infection [60]. Beta-amyrin has been found in leaf epicuticular waxes of oak [61] and its derivatives show antifungal activity [62]. Furthermore, among DEGs shared between Pr and Cr plants, 4 out of 25 genes encode leucine rich repeat (LRR) proteins. Homologs of the LRR proteins include chitin [62] and flagellin [63] receptor proteins in *Arabidopsis thaliana*. Three genes for flavonoid-modifying enzymes were also shared between the two comparisons. Flavonoids are structurally diverse secondary metabolites in plants and one of their important functions is defense against pathogens and herbivores [64]. Importantly phosphite changed expression of the DEGs in the same direction as innate resistant plants in relation to susceptible plants, which coincided with the acquisition of resistance to *Phytophthora* infection. This transcriptional response is in line with the proposed mechanism in which phosphite stimulates extant host defenses and increases the resistance of susceptible host plants to infection by *Phytophthora* [7,10,11]. 

### 4.4. Use of English Oak as a Reference for Tanoak RNA-Seq Analysis

Due to the unavailability of the genome sequence of tanoak, the English oak genome was used as a reference to analyze the tanoak transcriptome. Over 90% of tanoak sequencing reads were mapped to the English oak genome (Appendix A) and were subsequently used for data analysis. It should be noted that tanoak specific genes and genes diverged from homologs in English oak are not represented in this work. A genome sequence project of tanoak is currently underway, and use of the tanoak genome will inevitably improve the outcome of data analysis. Nevertheless, our present work provides a general picture of tanoak transcriptomes associated with innate resistance, phosphite-induced resistance, and phytophthora infection. It would be interesting to evaluate the conservation of gene regulations in phosphite-induced resistance in closely related host species such as oaks, chestnuts, and walnuts.

### 4.5. Phosphite-Induced Transcriptome Changes of P. ramorum

Several lines of evidence suggest that at high concentrations, phosphite directly inhibits the growth of *Phytophthora* species through direct toxicity [65]. The effect of phosphite on the transcriptome of *P. cinnamomi* grown on a culture medium has been investigated [66]. At 40 μg/mL of phosphite, *P. cinnamomi* showed a severe growth inhibition and lysis of the hyphal wall, while 32 genes were reported to be differentially expressed. There are several features shared with our in planta data set. For instance, multiple ABC transporters were detected in upregulated as well as downregulated gene sets of *P. cinnamomi* in response to phosphite. Likewise, several ABC transporters were also differentially expressed in *P. ramorum* in phosphite-treated plants. Homologs of two of the *P. ramorum* ABC transporters upregulated in phosphite-treated tanoak seedlings are found in plant pathogenic fungi *Magnaporthe oryzae* (*Abc3* gene) [67] and *Fusarium sambucinum* (*Gpabc1* gene) [68], and these fungal homologs participate in efflux of toxins. A vitamin B_6_ biosynthesis gene was detected in *P. cinnamomi*, which is consistent with our GO enrichment analysis where “pyridoxal phosphate biosynthetic process” (synonym: “active vitamin B_6_ biosynthesis”) was detected in the upregulated set in phosphite treated samples. In fungal plant pathogens, vitamin B_6_ [69] as well as glutathione S-transferase [70] (also upregulated in *P. ramorum* in phosphite-treated plant tissue) function as antioxidant stress protectors against reactive oxygen species. The observed upregulation of genes for antioxidant production in this research is consistent with the *P. palmivora* and *A. thaliana* pathosystem, in which phosphite-treated plants rapidly released superoxide [11]. Among the seven in planta transcriptomes of *P. ramorum*, two were from phosphite-treated resistant (Pr) plants. Neither of the two phosphite-treated T7 susceptible plant samples (Ps) yielded *P. ramorum* transcripts. In other words, both transcriptomes of *P. ramorum* from phosphite-treated plants represent those eventually contained by host defense systems. Lysis of hyphal wall and release of MAMPs will evidently further activate the host defense [1,4]. The direction of future work will be to understand phosphite-induced resistance at a high spatio-temporal resolution. This can be achieved by comparing *P. ramorum* invasion and progression in Cs and Pr plants through RNA-Seq while controlling developmental stages of the pathogen in planta by monitoring a fluorescent protein (GFP)-tagged strain of *P. ramorum* in the tanoak tissue.

## 5. Conclusions

Transcriptome analysis identified candidate genes involved in natural resistance to Sudden Oak Death, as well as genes possibly associated with phosphite-induced resistance. Sets of candidate genes for innate resistance and phosphite-induced resistance largely overlapped, and a large part of the overlapped genes implicated plant defense processes. Thus, our transcriptome analysis is in line with the hypothesis that phosphite increases the resistance of susceptible host species to *Phytophthora* infection. When transcriptomes of *P. ramorum* were compared in phosphite-treated and untreated plants, genes for membrane transporters and vitamin B_6_ biosynthesis were found active, which is consistent with direct toxicity of phosphite on the pathogen. We have shown the importance and power of dual RNA-Seq in plant-pathogen interactions as well as plant-pathogen-pesticide interactions. Perturbation of transcriptome due to interactions and enrichment of pathways or gene functions helped discern physiological changes of the host and pathogen. Differentially expressed genes associated with innate resistance and/or phosphite-induced resistance can be used to develop mRNA markers for screening and marker-assisted breeding of tanoak.

## Figures and Tables

**Figure 1 jof-07-00198-f001:**
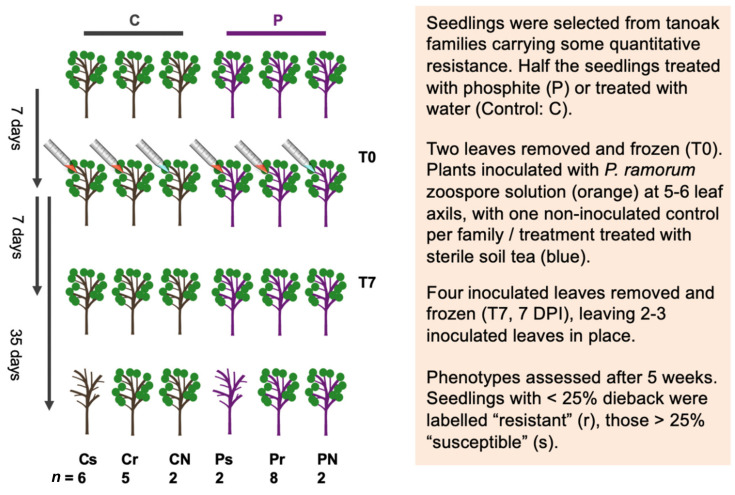
Experimental design. A total of 25 plants across 2 families were subjected to the experiment. Abbreviations are C: water control, P: phosphite treated, T0: time 0, T7: 7 days post inoculation, Cs: water control susceptible, Cr: water control resistant, CN: water control uninoculated, Ps: phosphite-treated susceptible, Pr: phosphite-treated resistant, PN: phosphite-treated uninoculated. Sample size (*n*) is indicated for each treatment group.

**Figure 2 jof-07-00198-f002:**
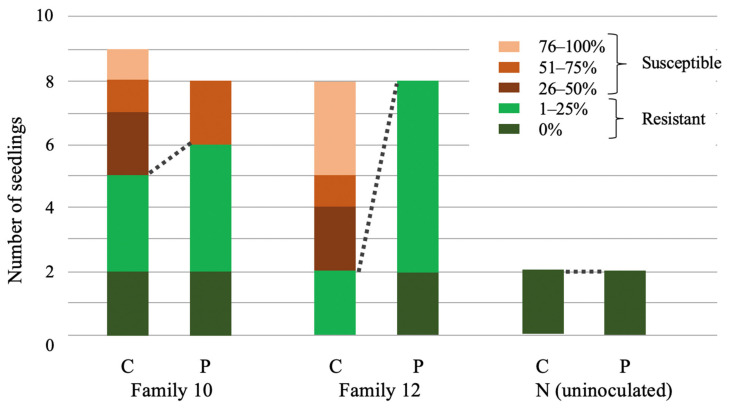
Effect of phosphite on reduction of tanoak dieback five weeks post inoculation. The y-axis shows the number of individual plants treated with either water (C) or phosphite (P) for each family. Dieback 0–25% is defined as resistant, and 26–100% susceptible. Samples with dieback between 26–50% were excluded from RNA-Seq analysis. None of the four uninoculated plants (N), treated with water or phosphite, showed disease symptoms. Dotted lines join portions of seedlings that showed resistance.

**Figure 3 jof-07-00198-f003:**
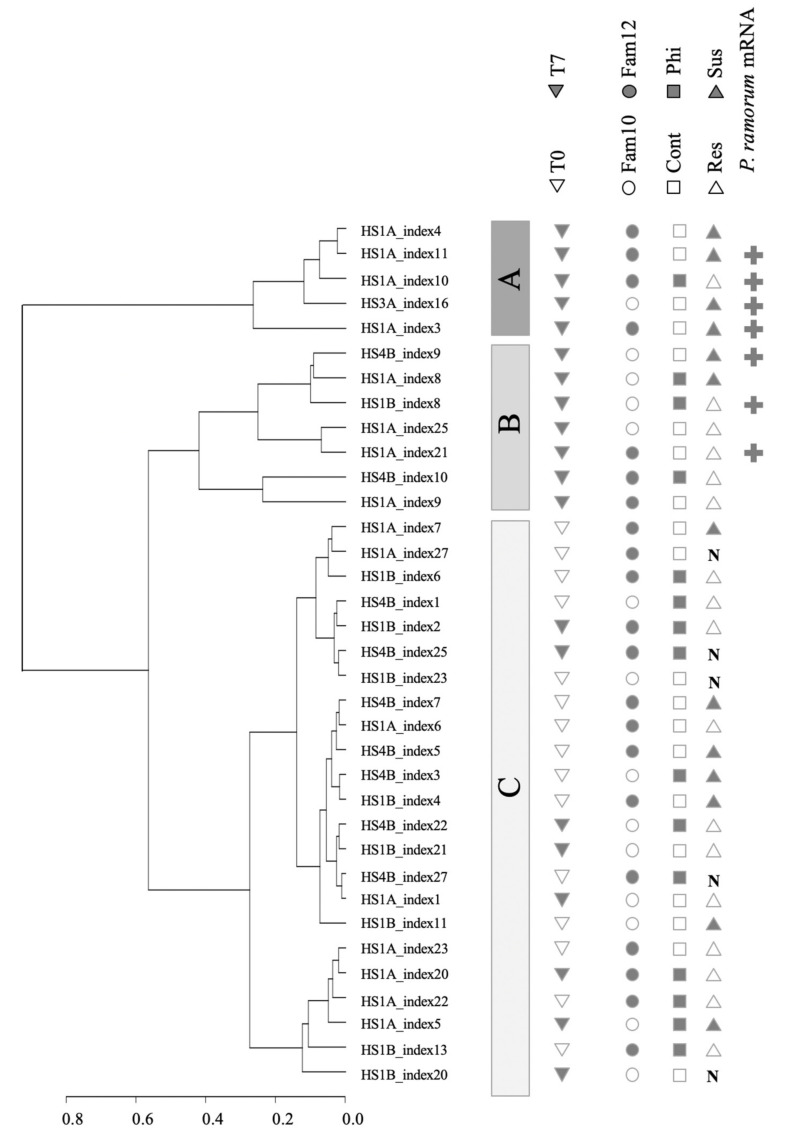
Hierarchical clustering of 35 tanoak transcriptomes showing three distinctive clusters A, B, and C. Sampling time (T0 or T7), tanoak family (Family 10 or 12), phosphite treatment (Phi: phosphite-treated and Cont: water control), disease phenotypes at 35 DPI (Res: resistant or Sus: susceptible), as well as occurrence of *P. ramorum* mRNA are indicated. N indicates non-inoculation control. Labels on terminal branches indicate IDs for cDNA libraries.

**Figure 4 jof-07-00198-f004:**
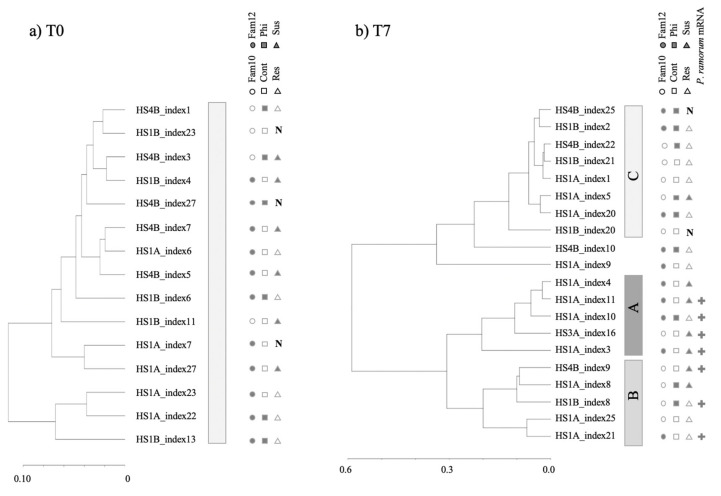
Hierarchical clustering of (**a**) 15 tanoak transcriptomes sampled at T0 and (**b**) 20 tanoak transcriptomes sampled at T7. Three clusters A, B, and C were defined in Figure 3. Tanoak family (Family 10 or 12), phosphite treatment (Phi: phosphite-treated and Cont: water control), disease phenotypes at 35 DPI (Res: resistant or Sus: susceptible), as well as occurrence of *P. ramorum* mRNA are indicated. N indicates non-inoculation control.

**Figure 5 jof-07-00198-f005:**
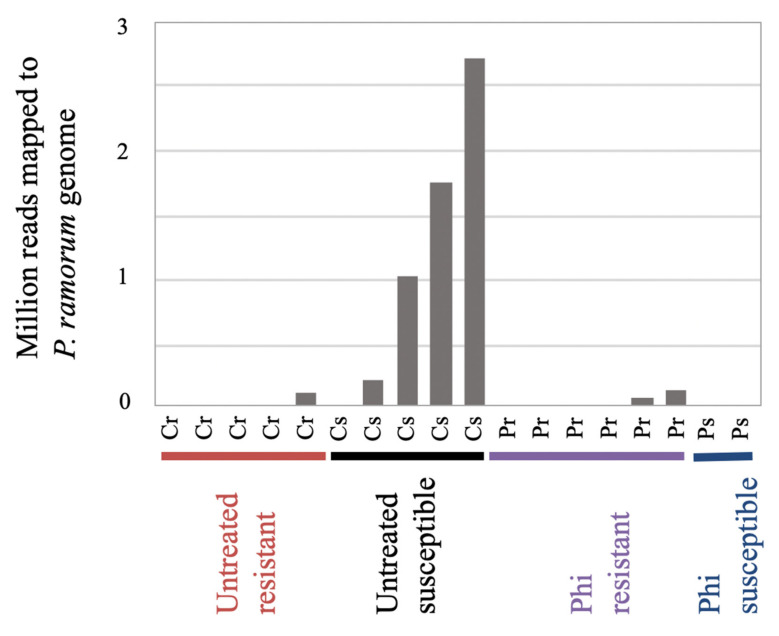
Number of reads from inoculated leaves at seven days post inoculation mapped to *P. ramorum* gene models [25] with Rsubread [34]. Samples are labeled as Untreated resistant (Cr), Untreated susceptible (Cs), Phosphite-treated resistant (Pr), Phosphite-treated susceptible (Ps).

**Table 1 jof-07-00198-t001:** Summary of deferentially expressed genes and enriched GO terms in group comparisons.

Comparison	Family	N v N ^1^	DE Up ^2^	GO and KEGG ^3^	DE Down ^2^	GO ^3^
			AB up		AB down	
AB and C clusters	F10 & F12	12 v 23	7178	226, 40	6265	143, 8
			Cr up		Cr down	
Cr and Cs plants at T0	F12	4 v 2	268	104, 4	198	6, 1
			P up		P down	
P and C plants at T0	F12	6 v 3	0	0, 0	0	0, 0
			Pr up		Pr down	
Pr and Cs plants at T0	F12	4 v 3	31	2, 0	16	0, 0

^1^ Numbers of seedlings used in the comparison; ^2^ Numbers of upregulated and downregulated deferentially expressed (DE) genes are shown (adjusted *p* < 0.05); ^3^ Number of enriched Gene Ontology (GO) Biological Process and Kyoto Encyclopedia of Genes and Genomes (KEGG) Pathways (adjusted *p* < 0.05); Abbreviations: C water control, P phosphite treated, T0 time 0, Cr water control resistant, Cs water control susceptible, Pr phosphite-treated resistant.

**Table 2 jof-07-00198-t002:** GO and KEGG pathway enrichment analysis on tanoak transcriptomes.

GO and KEGG ID ^1^	Cluster ^2^	Description	False Discovery Rate ^3^
map01130	AB_up	Biosynthesis of antibiotics	7.76 × 10^−17^
map00980	AB_up	Metabolism of xenobiotics by cytochrome P450	2.57 × 10^−11^
map00010	AB_up	Glycolysis/Gluconeogenesis	2.57 × 10^−9^
map00982	AB_up	Drug metabolism-cytochrome P450	2.88 × 10^−9^
map00480	AB_up	Glutathione metabolism	4.26 × 10^−9^
map00983	AB_up	Drug metabolism-other enzymes	6.02 × 10^−9^
map00520	AB_up	Amino sugar and nucleotide sugar metabolism	6.93 × 10^−9^
map04660	AB_up	T cell receptor signaling pathway	1.46 × 10^−6^
map00020	AB_up	Citrate cycle (TCA cycle)	1.56 × 10^−6^
map04658	AB_up	Th1 and Th2 cell differentiation	1.81 × 10^−5^
map00400	AB_up	Phenylalanine, tyrosine and tryptophan biosynthesis	6.29 × 10^−5^
map00230	AB_up	Purine metabolism	6.48 × 10^−5^
map00940	AB_up	Phenylpropanoid biosynthesis	6.63 × 10^−5^
map00730	AB_up	Thiamine metabolism	7.04 × 10^−5^
map00830	AB_up	Retinol metabolism	7.92 × 10^−5^
map00720	AB_up	Carbon fixation pathways in prokaryotes	1.80 × 10^−4^
map00071	AB_up	Fatty acid degradation	2.50 × 10^−4^
map00592	AB_up	alpha-Linolenic acid metabolism	2.89 × 10^−4^
map00625	AB_up	Chloroalkane and chloroalkene degradation	1.29 × 10^−3^
map00620	AB_up	Pyruvate metabolism	1.30 × 10^−3^
map00680	AB_up	Methane metabolism	1.37 × 10^−3^
map00640	AB_up	Propanoate metabolism	1.41 × 10^−3^
map00760	AB_up	Nicotinate and nicotinamide metabolism	2.01 × 10^−3^
map00626	AB_up	Naphthalene degradation	2.58 × 10^−3^
map00260	AB_up	Glycine, serine and threonine metabolism	3.12 × 10^−3^
map00270	AB_up	Cysteine and methionine metabolism	3.48 × 10^−3^
map00040	AB_up	Pentose and glucuronate interconversions	6.21 × 10^−3^
map00350	AB_up	Tyrosine metabolism	6.84 × 10^−3^
map00051	AB_up	Fructose and mannose metabolism	9.69 × 10^−3^
GO:0016567	AB_up	protein ubiquitination	4.16 × 10^−9^
GO:0010951	AB_up	negative regulation of endopeptidase activity	5.62 × 10^−8^
GO:0006888	AB_up	endoplasmic reticulum to Golgi vesicle-mediated transport	1.02 × 10^−4^
GO:0006749	AB_up	glutathione metabolic process	1.68 × 10^−4^
GO:0006032	AB_up	chitin catabolic process	3.14 × 10^−4^
GO:0032482	AB_up	Rab protein signal transduction	3.14 × 10^−4^
GO:0006468	AB_up	protein phosphorylation	3.99 × 10^−4^
GO:0031640	AB_up	killing of cells of other organism	4.86 × 10^−4^
GO:0009435	AB_up	NAD biosynthetic process	1.02 × 10^−3^
GO:0010200	AB_up	response to chitin	1.57 × 10^−3^
GO:0006099	AB_up	tricarboxylic acid cycle	1.57 × 10^−3^
GO:0016998	AB_up	cell wall macromolecule catabolic process	3.51 × 10^−3^
GO:0006614	AB_up	SRP-dependent cotranslational protein targeting to membrane	3.51 × 10^−3^
GO:0009694	AB_up	jasmonic acid metabolic process	3.76 × 10^−3^
GO:0006096	AB_up	glycolytic process	4.47 × 10^−3^
GO:0006457	AB_up	protein folding	6.42 × 10^−3^
GO:0061025	AB_up	membrane fusion	6.54 × 10^−3^
GO:0002181	AB_up	cytoplasmic translation	8.26 × 10^−3^
GO:0009423	AB_up	chorismate biosynthetic process	9.43 × 10^−3^
GO:0009873	AB_up	ethylene-activated signaling pathway	9.66 × 10^−3^
map00860	AB_down	Porphyrin and chlorophyll metabolism	4.23 × 10^−5^
map00670	AB_down	One carbon pool by folate	1.14 × 10^−3^
map00970	AB_down	Aminoacyl-tRNA biosynthesis	2.79 × 10^−3^
GO:0010088	AB_down	phloem development	5.04 × 10^−5^
GO:0009768	AB_down	photosynthesis, light harvesting in photosystem I	1.06 × 10^−4^
GO:0018298	AB_down	protein-chromophore linkage	1.22 × 10^−4^
GO:0006298	AB_down	mismatch repair	1.78 × 10^−3^
GO:0010206	AB_down	photosystem II repair	1.95 × 10^−3^
GO:0006418	AB_down	tRNA aminoacylation for protein translation	2.36 × 10^−3^
GO:0009234	AB_down	menaquinone biosynthetic process	3.43 × 10^−3^
GO:0045037	AB_down	protein import into chloroplast stroma	3.43 × 10^−3^

^1^ IDs for Enriched Gene Ontology (GO) terms and Kyoto Encyclopedia of Gene and Genomes (KEGG) pathways were listed. GO (Biological Process only) was reduced to most specific terms.^2^ Hierarchical clusters of the DEGs and direction of gene expressions (up or down) are indicated. ^3^ False discovery rate was used as correction for multiple tests [45]. For both KEGG and GO, adjusted *p*-value < 0.01 are shown.

**Table 3 jof-07-00198-t003:** GO enrichment analysis of deferentially expressed genes (DEGs) among innate resistant Family 12 plants.

GO and KEGG ID ^1^	DEG Category ^2^	Description	False Discovery Rate
**Cr v. Cs**			
GO:0010951	Cr_up	negative regulation of endopeptidase activity	3.15 × 10^−9^
GO:0055114	Cr_up	oxidation-reduction process	1.35 × 10^−6^
GO:0042744	Cr_up	hydrogen peroxide catabolic process	3.19 × 10^−5^
GO:0016114	Cr_up	terpenoid biosynthetic process	2.08 × 10^−4^
GO:0098869	Cr_up	cellular oxidant detoxification	3.11 × 10^−4^
GO:0006979	Cr_up	response to oxidative stress	3.64 × 10^−4^
GO:0009423	Cr_up	chorismate biosynthetic process	1.17 × 10^−2^
GO:0031408	Cr_up	oxylipin biosynthetic process	1.17 × 10^−2^
GO:0046129	Cr_up	purine ribonucleoside biosynthetic process	1.73 × 10^−2^
GO:0009072	Cr_up	aromatic amino acid family metabolic process	2.47 × 10^−2^
GO:0006833	Cr_up	water transport	3.06 × 10^−2^
GO:0008654	Cr_up	phospholipid biosynthetic process	3.34 × 10^−2^
GO:0030245	Cr_up	cellulose catabolic process	4.40 × 10^−2^
GO:0009742	Cr_up	brassinosteroid mediated signaling pathway	4.40 × 10^−2^
GO:0046130	Cr_up	purine ribonucleoside catabolic process	4.45 × 10^−2^
GO:0009693	Cr_up	ethylene biosynthetic process	4.45 × 10^−2^
GO:0010087	Cr_up	phloem or xylem histogenesis	4.67 × 10^−2^
map00940	Cr_up	Phenylpropanoid biosynthesis	8.47 × 10^−10^
map01130	Cr_up	Biosynthesis of antibiotics	2.10 × 10^−6^
map00270	Cr_up	Cysteine and methionine metabolism	6.46 × 10^−3^
map00500	Cr_up	Starch and sucrose metabolism	4.97 × 10^−2^
GO:0006417	Cr_down	regulation of translation	4.33 × 10^−2^
GO:0006075	Cr_down	(1->3)-beta-d-glucan biosynthetic process	4.33 × 10^−2^
GO:0009682	Cr_down	induced systemic resistance	4.33 × 10^−2^
map00500	Cr_down	Starch and sucrose metabolism	1.13 × 10^−4^
**Cr v. Cs & Cs v. Pr overlap**			
GO:0010088	Cr_up & Pr_up	phloem development	1.26 × 10^−2^
GO:0016104	Cr_up & Pr_up	triterpenoid biosynthetic process	4.50 × 10^−2^

^1^ IDs for Enriched Gene Ontology (GO) terms and Kyoto Encyclopedia of Gene and Genomes (KEGG) pathways were listed. GO terms were reduced to most specific, and only GO terms in Biological Process category were shown; ^2^ Upregulated or downregulated in Cr or Cr and Pr plants; Abbreviations: Cs water control susceptible, Cr water control resistant, Pr: phosphite-treated resistant.

**Table 4 jof-07-00198-t004:** Deferentially expressed genes (DEGs) shared within tree family 12 in the comparisons of untreated susceptible trees (Cs) and phosphite-treated resistant trees (Ps) as well as untreated susceptible trees (Cs) and untreated resistant trees (Cr) before inoculation.

DEG Category ^1^	Cs and Pr at T0 (Phi-Induced Resistance)	Cs and Cr at T0 (Innate Resistance)	Overlap
Cs_up	16	198	3
Cs_down	31	268	22

^1^ Differentially expressed genes are categorized to upregulated (Cs_up) and downregulated (Cs_down) in uninoculated susceptible trees.

## Data Availability

Not applicable.

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
