# Peer review of "Innate Resistance and Phosphite Treatment Affect Both the Pathogen’s and Host’s Transcriptomes in the Tanoak-Phytophthora ramorum Pathosystem"

_jof, 2021, doi:10.3390/jof7030198_

Round 1
Reviewer 1 Report
This is a clearly written and interesting study aimed at elucidating the mechanisms of phopshites upon sudden oak death: both the pathogen and the host plant. It uses a transcriptomics approach. Notwithstanding the usual caveats about inferences based only on transcriptomics and not followed-up with experiment-based hypothesis testing, this work is competently performed and provides some interesting insights. The experimental design presents an interesting and clever solution to the problem of not knowing in advance the resistance phenotype of the host.
I do have some suggestions for improvement, most of which are quite trivial, and a few comments that the authors may wish to consider.
- Line 18. The reader is less interested in learning that you designed an experiment and more interested in the fact that you executed the experiment and what the results were.
- Line 19. "treatment-individuals" sounds a bit weird. Do you simply mean treated individuals?
- Lines 27 - 28 and also the Discussion section. The authors commit a logical fallacy when they say that because detoxification genes are upregulated by phosphite then phosphite must be toxic. This is an example of "affirming the consequent".
- Line 74: "is" should be "are".
- Line 75: I'm not really sure what is meant by a "de novo transcriptome". A transcriptome is the set of transcription products, i.e. the RNA. In what sense is this RNA de novo? Probably the authors talking about a reference transcriptome sequence that has been assembled de novo?
- Line 96: you need a semicolon (or full stop) before "however".
- Line 213: which version/release of SAMtools was used?
- You need to cite Li H, Handsaker B, Wysoker A, Fennell T, Ruan J, Homer N, Marth G, Abecasis G, Durbin R; 1000 Genome Project Data Processing Subgroup. The Sequence Alignment/Map format and SAMtools. Bioinformatics. 2009 Aug 15;25(16):2078-9. doi: 10.1093/bioinformatics/btp352. Epub 2009 Jun 8. PMID: 19505943; PMCID: PMC2723002.
- Line 220: Gene Ontology not Gene ontology.
- Lines 227 and 243: is it DEseq2 or DESeq2?
- Line 219: this sentence is poorly written. It should be something like "the most closely related species whose genome has been sequenced".
- Line 228: it is not the parameters that are default; it is the values of those parameters that are default.
- Line 252: you should cite R Core Team (2014). R: A language and environment for statistical computing. R Foundation for Statistical
Computing, Vienna, Austria. URL http://www.R-project.org/ - Line 252: how was this Fisher test implemented? Was it using native R language? Or some library that should be cited? I notice there is a supplementary document about Fisher test, but this is not mentioned in this section of the main text.
- Line 252: why is R 3.3.3 mentioned for the first time here yet several R-based analyses were described on the previous page without mentioning R at all?
- Line 258: I am a little confused about the organisation of the sequence data. The tanoak cDNA was not physically separated from the Phytopthora cDNA was it? In other words, a single BioSample (single sequencing library?) was sequenced that includes a metatranscriptome of plant plus pathogen. The sequence data should not be separated into separate runs and separate BioSamples if they came from the same physical library or same biological sample. Th raw data should be submitted as-is, allowing other researchers to apply their own bioinformatic separation of the data. For reproducibility, raw data should always be raw!
- Line 287: chloroplasts not Chloroplasts.
- Line 338: do you really mean 'homologous' (i.e sharing a common evolutionary origin) or do you simply mean 'similar'?
Author Response
Reviewer 1
Comments and Suggestions for Authors
This is a clearly written and interesting study aimed at elucidating the mechanisms of phopshites upon sudden oak death: both the pathogen and the host plant. It uses a transcriptomics approach. Notwithstanding the usual caveats about inferences based only on transcriptomics and not followed-up with experiment-based hypothesis testing, this work is competently performed and provides some interesting insights. The experimental design presents an interesting and clever solution to the problem of not knowing in advance the resistance phenotype of the host.
I do have some suggestions for improvement, most of which are quite trivial, and a few comments that the authors may wish to consider.
Line 18. The reader is less interested in learning that you designed an experiment and more interested in the fact that you executed the experiment and what the results were.
<Our Response> "designed" was changed to "conducted"
Line 19. "treatment-individuals" sounds a bit weird. Do you simply mean treated individuals?
<Our Response> It was changed to "treatment group individuals".
Lines 27 - 28 and also the Discussion section. The authors commit a logical fallacy when they say that because detoxification genes are upregulated by phosphite then phosphite must be toxic. This is an example of "affirming the consequent".
<Our Response> In Abstract, we removed "indicating the toxic effect of phosphite". In Conclusion, we added "which is consistent with" to tone down the statement. In Discussion, toxicity present in related organisms are only discussed.
Line 74: "is" should be "are".
<Our Response> corrected. Thank you.
Line 75: I'm not really sure what is meant by a "de novo transcriptome". A transcriptome is the set of transcription products, i.e. the RNA. In what sense is this RNA de novo? Probably the authors talking about a reference transcriptome sequence that has been assembled de novo?
<Our Response> We inserted its definition in parentheses as "a de novo assembled transcriptome (a reconstructed transcriptome from RNA-Seq experiments)"
Line 96: you need a semicolon (or full stop) before "however".
<Our Response> semicolon added. (now L123)
Line 213: which version/release of SAMtools was used?
<Our Response> version 1.9 added.
You need to cite Li H, Handsaker B, Wysoker A, Fennell T, Ruan J, Homer N, Marth G, Abecasis G, Durbin R; 1000 Genome Project Data Processing Subgroup. The Sequence Alignment/Map format and SAMtools. Bioinformatics. 2009 Aug 15;25(16):2078-9. doi: 10.1093/bioinformatics/btp352. Epub 2009 Jun 8. PMID: 19505943; PMCID: PMC2723002.
<Our Response> Li et al., added
Line 220: Gene Ontology not Gene ontology.
<Our Response> corrected.
Lines 227 and 243: is it DEseq2 or DESeq2?
<Our Response> Corrected to DESeq2.
Line 219: this sentence is poorly written. It should be something like "the most closely related species whose genome has been sequenced".
<Our Response> Corrected as suggested, "The genome of English oak Quercus robur, the most closely related species whose genome has been sequenced, was used as a reference genome".
Line 228: it is not the parameters that are default; it is the values of those parameters that are default.
<Our Response> Corrected to "The default values of the parameters"
Line 252: you should cite R Core Team (2014). R: A language and environment for statistical computing. R Foundation for Statistical Computing, Vienna, Austria. URL http://www.R-project.org/
<Our Response> The reference replaced as suggested.
Line 252: how was this Fisher test implemented? Was it using native R language? Or some library that should be cited? I notice there is a supplementary document about Fisher test, but this is not mentioned in this section of the main text.
<Our Response> The function used was given as "using Fisher’s exact test function fisher.test()"
Line 252: why is R 3.3.3 mentioned for the first time here yet several R-based analyses were described on the previous page without mentioning R at all?
<Our Response> The citation was moved to Line 210 (now L316), where R is first mentioned.
Line 258: I am a little confused about the organisation of the sequence data. The tanoak cDNA was not physically separated from the Phytopthora cDNA was it? In other words, a single BioSample (single sequencing library?) was sequenced that includes a metatranscriptome of plant plus pathogen. The sequence data should not be separated into separate runs and separate BioSamples if they came from the same physical library or same biological sample. Th raw data should be submitted as-is, allowing other researchers to apply their own bioinformatic separation of the data. For reproducibility, raw data should always be raw!
<Our Response> The raw data (before subtraction) aligned to tanoak cDNA as well as to Phytophthora genomic DNA including unaligned reads were submitted to NCBI. We clarified as " BAM files for the original Illumina RNA sequencing data aligned to the de novo tanoak transcriptome library and the same data aligned to P. ramorum reference genome were deposited in the NCBI Sequence Read Archive under study accessions SRP157197 and SRP157863, respectively.
Line 287: chloroplasts not Chloroplasts.
<Our Response> Corrected.
Line 338: do you really mean 'homologous' (i.e sharing a common evolutionary origin) or do you simply mean 'similar'?
<Our Response> Changed to "highly similar" as we do not know whether similar sequences share a common ancestry.
Reviewer 2 Report
Kasuga and colleagues provide a nice study to investigate responses of tanoak to P. ramorum infection and phosphite treatment. The MS details a lot of work, and has a lot of nice results which help us understand how phosphite helps plants resist Phytophthora infection. The study was conducted on an important host, and important pathogen. The methods used were robust, and detailed analysis conducted.
While I propose the paper is a valuable contribution and should be published in JoF, I feel it needs quite a few corrections before it is acceptable. In many areas the paper is more like draft rather than a version ready for publication. There are a few instances of repetition that should be addressed to shorten the paper.
It is a shame you did not examine for changes in transcriptome before 7 Days past treatment, as you said you might have seem more of the priming effect. Can you explain why you picked 7 days?
I have made detailed comments on the PDF, but repeat a few here too.
In a few places you discuss innate resistance, but don’t give your definition of it. Reading this MS innate seems like it covers all kinds of resistance that is not due to phosphite, but I think there is more to it. Can you explain this early in the intro
The tables especially need a lot of work, as they are not readily understandable without reference to the text.
The methods section could use a bit of tidying, as there is some repetition and also certain areas would more logically be placed closer together.
Much of the bioinformatics sections are provided without sufficient explanation. There are a lot of abbreviations used that are not always explained.
There is some repetition between the results and methods sections.
In some areas of the results there are references cited. I don’t know what the JoF style is, but usually for me when references are needed I consider the sentence to be better suited in the discussion.

Author Response
Reviewer 2
Comments and Suggestions for Authors
Kasuga and colleagues provide a nice study to investigate responses of tanoak to P. ramorum infection and phosphite treatment. The MS details a lot of work, and has a lot of nice results which help us understand how phosphite helps plants resist Phytophthora infection. The study was conducted on an important host, and important pathogen. The methods used were robust, and detailed analysis conducted.
While I propose the paper is a valuable contribution and should be published in JoF, I feel it needs quite a few corrections before it is acceptable. In many areas the paper is more like draft rather than a version ready for publication. There are a few instances of repetition that should be addressed to shorten the paper.
It is a shame you did not examine for changes in transcriptome before 7 Days past treatment, as you said you might have seem more of the priming effect. Can you explain why you picked 7 days?
<Our Response> (L392) We added why we waited for 7 days "Our previous work has showed that efficacy of the systemic fungicide is consistently seen seven days post treatment [47]". The priming effect better studies in a shorter time frame is of great interest; however, that experiment would be more successful if performed when clonal propagation of tanoak is established that would allow us to conduct fine time-scale analysis of gene regulation.
I have made detailed comments on the PDF, but repeat a few here too.
In a few places you discuss innate resistance, but don’t give your definition of it. Reading this MS innate seems like it covers all kinds of resistance that is not due to phosphite, but I think there is more to it. Can you explain this early in the intro
<Our Response> We added definition in Abstract as well as Introduction:
"(resistance displayed by untreated tanoak)"
" ... approximately 20% of offspring demonstrated a resistant phenotype (dieback 25% or less) [21]. Hereafter, the resistance phenotype of phosphite-untreated tanoak is defined as "innate resistance" to distinguish it from phosphite-induced resistance."
The tables especially need a lot of work, as they are not readily understandable without reference to the text.
The methods section could use a bit of tidying, as there is some repetition and also certain areas would more logically be placed closer together.
Much of the bioinformatics sections are provided without sufficient explanation. There are a lot of abbreviations used that are not always explained.
There is some repetition between the results and methods sections.
In some areas of the results there are references cited. I don’t know what the JoF style is, but usually for me when references are needed I consider the sentence to be better suited in the discussion.
Submission Date
15 January 2021
Date of this review
06 Feb 2021 18:34:05
detailed comments written in MS
L37: reference for control of things other than P. ramorum?
<Our Response> Two seminal papers; by Guest et al., and by Hardy et al., were added.
L40: can you add a sentence to explain what phosphite compounds are for the reader. what is their normal use, when were they first applied to plant protection/disease management. At the moment you are assuming the reader is already fully aware of what they are.
<Our Response> A sentence added to introduce phosphite to the reader: "Phosphites, salts or esters of phosphonic acid, are systemic compounds first shown to be highly effective against diseases caused by oomycetes in the 1970s (reviewed in [1]) and have since been used widely as fungicides in horticulture and natural ecosystems [2]".
L56: how can something have phylogenetic importance/? nothing is really important phylogenetically speaking. are you saying that genetic diversity of tanoak is important. why is this important
<Our Response> There is only one species in the genus Notholithocarpus, however, we do agree that the meaning of "phylogenetic importance" is obscure. We, therefore, deleted the two words.
L57: i dont think this sentence follows from the previous. Commercial forestry is normally interested in timber. if tanoak isnt a useful time species then it probably wont be valued by commercial forestry.
<Our Response> Although tanoaks are ecologically and culturally important, they have little commercial value. We clarified as: " While the ecological and cultural importance of tanoaks is well established [23], owing primarily to their low commercial value [1], tanoaks have not been widely propagated for forestry.
L75: pathosystem
<Our Response> "considerable genomic resources available in this pathosystem" added
L82: did you identify these in this study, or in a different study. if its this study, then maybe this info should be in methods. if it was another study please give a reference
<Our Response> The finding was described before and the reference "Hayden et al., 2013" was added.
L85: move ln 81 - 90 to methods section
<Our Response> Lines 81-90 were moved to Material and Methods, Project Overview.
L91: here and ln 99-101 you make hypotheses. can you birng both together, or at least move them to the same paragraph
<Our Response> Two hypotheses were combined in one paragraph starting at L118.
L94: i think you need to define what you class as innate resistance? is it the ability of the plant to resist infection, or contain/destroy pathogen once it has infected. or is it both. or do you just mean the ability of plants to not show disease symptoms which is not related to phosphite induced resistance.
<Our Response> Definition added as "Hereafter, the resistance phenotype (dieback 25% or less) of phosphite-untreated tanoak is defined as "innate resistance" to distinguish it from phosphite-induced resistance."
L105: there is not data on the first experiment? only fam 10 and 12 are ever mentioned. could you change to just deal with 1 experiment and forget about the preliminary screening. or say somewhere (data not shown) for th other two families.
<Our Response> The preliminary screening was removed from Project overview.
L111: do you mean experiment here instead of trial?, in reference to the 2nd experiment being carried out on two half sib families
<Our Response> The preliminary trial has been removed from this manuscript. The experimental design for gene expression analysis is shown here and we changed the section subtitle to: "Experimental design for gene expression analysis"
L119: should this be "sibling", "sapling" or actually "sampling"?
<Our Response> It was "sapling". We had also used "seedlings" but to be consistent, the word sapling was replaced with seedling throughout the manuscript.
Fig1: this gives the impression that you used a water control for your uninoculated? or did you not apply any phytophthora control to your uninoculated?
<Our Response> We did use water control for uninoculated, which is stated in the legend, "CN: water control uninoculated" and "PN: phosphite-treated uninoculated". We added orange color to zoospore suspension whereas left blue for water control to contrasts the treatments.
L124: should this be 24?
<Our Response> As shown in Fig. 1, N= 6 + 5 + 2 + 2 + 8 + 2, the total number of samples is 25.
L134: by "as well as" do you mean in combination, or that both are effective seperately. if its the latter, then delete the bit about injections is irrelevant
<Our Response> To make it clear, we deleted "injections of phosphites as well as".
L146: merge this with the next section to make "Innoculation and leaf harvest
<Our Response> Two sections, "Isolate and inoculum preparation" and "Plant inoculation and leaf harvest" merged as suggested.
L154: the sealed vessel placed in ice.. IM confused as to whether the agar plugs were poured into ice water, or the petri dished were placed into a sealed container and then the contrainer placed in ice water.
<Our Response> To make it clear, we rewrote the experimental procedure for zoospore release: "Zoospore release from sporangia was induced by cold shocking the cultures as follows. The mycelial squares incubated in multiple petri plates were consolidated into a single vessel before being placed in ice water for 30 min. After that the mycelial squares were further incubated at room temperature for one hour."
L221: what is KEGG
<Our Response> (L332) When the acronym is used for the first time, we spelled out as: "Kyoto Encyclopedia of Genes and Genomes (KEGG)"
L273: based on what statistical test?
<Our Response> Thanks for pointing out. We now show only families 10 and 12 so the statistics is not relevant any longer. We deleted "accompanied by a marginally significant difference in innate susceptibility among families"
L274: is this the point where you exclude the other families and go with 10 and 12? can you saw something to this effect, like families X and Y were excluded from further analysis.
<Our Response> We deleted information for family 5 and 16, so we do not need to mention here.
Fig2: what is this dotted line?
<Our Response> We added in the Figure 2 legend: "Dotted lines join portions of seedlings that showed resistance."
L289: from here to 291 is methods not results
<Our Response> L285-296 were deleted as the information described in Materials and Methods.
L293: from here to 296 is discussion. no need to have references in the results section
<Our Response> We deleted as these two lines were not informative.
L297: from here to 298 is methods section
<Our Response> The sentence was deleted as it is described in Methods.
Fig3: add note to legend explaining what cont, phi, res, sus = mean
<Our Response> We added explanations for abbreviated words as: "Cont: Control, Phi: phosphite-treated, Res: resistant, Sus: susceptible".
Fig4: why dont you identify the 3 groups in T0?
<Our Response> A and B clusters were associated with infection, therefore none of T0 samples (before inoculation) were found in A or B. We added "Fig. 4a" in L485 to clarify: "all the T0 samples (resistant or susceptible, Fig. 4a)"
L328: i dont think enriched is a good term here. clsuter a was dominated by T7 and susceptible phenotypes.
<Our Response> We replaced "enriched" with "contained primarily" as: "Cluster A contained primarily T7 samples with susceptible phenotypes"
L329: why not just use normal convention of p<0.05" and what statistical test is this
<Our Response> Fisher's exact test, added. JoF does not specify how to report a P-value. We follow a convention below and changed values accordingly.
https://scc.ms.unimelb.edu.au/resources-list/understanding-empem-values/report-on-p-values
- If you are reporting P-values in an academic paper or thesis, it's good practice to report the actual value to three decimal places.
- If the P-value is very small, common practice is to report it as P < 0.001.
Table1: this table is very rough, and needs some additional explanation to make it easily understood, without having to refer to the text. What is "code"?
explain DEG, DE, Cr, Pr, AB, GO, KEGG
PLease take another look at this table
<Our Response>
Code was changed to "Comparison" and gave comparisons details.
DEG, DE, Cr, Pr, AB, GO, KEGG were all spelled out at the footnote.
L360: were they down in AB or up in C?
if they were down in AB then dont say the C part
<Our Response>Added the information (underlined) as: "highlighting upregulation of genes involved in defense (high in AB)"
L362: in which? AB or C
<Our Response> Shown as " enriched in cluster AB"
L362: why is this noteworthy? it seems to go against the idea tat C was the resistant phenotype, unless it was also up in C?`
<Our Response> Cluster C contains uninfected plants as well as inoculated but resistant plants. This implicates that gene expression of Cluster C is that for uninfected plants. Inoculated plants in this cluster are likely those that had already fended off the pathogen. As stated in L364, "Downregulation of photosynthesis genes and upregulation of genes for energy generation are hallmarks of plant immune processes", but it is not necessary associated with the resistant phenotype. To emphasize, we added "plant defense" in L534 as: "transcriptomes in cluster AB represent infection and plant defense"
Table2: explain what GO, DEG KEGG are
this table also looks unfinished. Can you add more explanation to what your labels/text mean. I guess AB down is genes that were down regulasted in your AB cluster?. this isnt explained.
what is FDR? is it a measure of biological activity, or is it a statistical number
<Our Response> GO and KEGG spelled out at the footnote as "GO: Gene Ontology (GO), KEGG: Kyoto Encyclopedia of Genes and Genomes", and FDR spelled out.
L385: again, why not the normal convention P<0.005)
<Our Response> As wrote previously, we decided to use "P<" only when P is smaller than 0.001.
Fig5: explain Phi, Pr, Cr CS Ps
<Our Response> In Fig. 5 legend, we added "Samples are labeled as Untreated resistant (Cr), Untreated susceptible (Cs), Phosphite-treated resistant (Pr), Phosphite-treated susceptible (Ps)".
T7
<Our Response> Spelled out as "seven days post inoculation"
L407: this sounds like methods
<Our Response> Rephrased it as below and summarizes the main idea of the paragraph: "Untreated susceptible trees (Control susceptible: Cs) and untreated resistant trees (Control resistant: Cr) before inoculation were compared to search for a transcriptome signature for innate resistance (Cs and Cr at T0 in Table 1, Table S4).
Table3:
as with previous tables, please spend some time making sure a reader can understand all this table without searching through the main text
<Our Response> Made the table header easier for readers to understand. Also listed abbreviations at the footnote and explained the table in details.
also say which plants you included, e.g exclude Fam 10
<Our Response> Family 12 added to the table title.
L423: here to 425 is discussion
<Our response> deleted as it is repetitive: "Exogenously applied phosphite is rapidly absorbed, translocated systemically, and primes susceptible hosts for a more rapid and intense defense response following recognition of a pathogen [53,54]."
L428: what does this sentence mean? innate resistance? where is this coming from so late in the MS? is innate meaning resistance without phosphite?
<Our response> It is not known if phosphite-treated resistant plants would display resistance without phosphite treatment due to unavailability of clonal propagation.
Definition of innate resistance is added to introduction as: "Hereafter, the resistance phenotype of phosphite-untreated tanoak is defined as "innate resistance" to distinguish it from phosphite-induced resistance."
L430: what is innate susceptible?
This paragraph needs a careful review and please consider whether its really necessary to confuse matters with innate terminology
from ln 430 to 433 are fine
<Our response> We deleted "susceptible" and rephrased it as "the most of Pr trees from Family 12 were unlikely to be innate-resistant to P. ramorum".
(L643) We need to explain readers why susceptible untreated plants and phosphite-treated resistant plants were compared, and also its caveat that phosphite-treated resistant plants could be resistant without phosphite treatment. To make it easier to follow, we have removed L423 to 425 in response to the previous comment.
L437: from 435 to 439 are really difficult to understand. you are trying to point out that certain changes in genes occured in some different groups after treatments i think. can you try to explain it better. sorry i cant be of any more help in suggesting wording, as ive tried a few times but cant grasp the point
<Our response> L647 We shortened and attempted to explain more concisely:
"It was found that over half of DEGs (22 out of 31) in the Cs Pr comparison are also DEGs in the Cs Cr comparison (Table 4). In other words, the changes in gene expression patterns observed in Cs plants following phosphite treatment (i.e. DEGs between Cs and Pr) are positively correlated with the difference in gene expression between Cs and Cr plants."
L444: in the end, what is the point of this paragraph? can you add a sentence to conclude the importance of this paragraph?
<Our response> A sentence added, which will be revisited in discussion: " These proteins have been implicated in active defense."
Table4: please tidy up this table
<Our response> We added more explanation as in the previous tables.
L461: strange to see references in results section. please move from here to ln 463 to discussion
<Our response> L461 to 463 was moved to discussion.
Table5: what is the point of this table if it only shows 1 row. this info could as easily be given in a sentence
<Our response> We removed Table 5 and added p-value to the text:
L709 "GO enrichment analysis identified “pyridoxal phosphate (Vitamin B6) biosynthesis process” among phosphite upregulated genes (false discovery rate corrected p=0.015)."
Discussion
L476: this sentence is confusing, and could do with a stop after families. Alternatively, what does this sentence add to the discussion.
<Our response> L475-477 were removed as we mainly analyzed only one family.
<Our response> L493 to L495 removed as suggested.
L505: did refs 11 and 47 really say that innate resistance could not prevent infection, and could not help clear the pathogen?. this sentence is very complicated, as you are talking about innate and phosphte tolerance, acting at point of inoculation and distant to point of infection, and also saying that your results were different to refs 11 and 47. can you check this sentence carefully and see if it is accurate
<Our response> Refs 11 and 47 focus on the early response of phosphite treated plants and do not address how plants deal with more established pathogens. We deleted L500-512 as we realized it was not sufficient finding for discussion.
L509: is this a suggestion for future research? it comes a bit out of the blue. Could you preface it by saying something like "Future research to..."
<Our response> We deleted L509 as the line was not informative.
L519: this seems wrong, should it be multiple genes each with small effects
<Our response> Corrected to " multiple and diverse classes of genes each with small effects "
L522: " QDR genes downstream to the event of MAMPs perception would not probably be detectable through the transcriptome comparison of uninoculated resistant and susceptible plants. " why? reference
<Our response> In response to MAMPs, plants activate downstream genes via protein kinase signaling networks. Therefore, QDR downstream to MAMPs perception may be at a basal gene expression level. We rewrote the sentence to clarify as: "QDR genes downstream to the event of MAMPs perception could be at a basal low expression state before inoculation, therefore they might not be detectable through the transcriptome comparison of uninoculated resistant and susceptible plants."
L528: "this GO term"?
<Our response> Corrected to "Enrichment of this GO term"
L587: can you merge your earlier future hypothesis on GFP with this one to have a shorter sentence covering it all
<Our response> We deleted the first hypothesis. Now we have L888 "The direction of future work will be to understand phosphite-induced resistance at a high spatio-temporal resolution. This can be achieved by comparing P. ramorum invasion and progression in Cs and Pr plants through RNA-Seq while controlling developmental stages of the pathogen in planta by monitoring a fluorescent protein (GFP)-tagged strain of P. ramorum in the tanoak tissue."
L602: is phosphite actually classed as a pesticide in USA? it is not in europe, it is a grey area that doesnt claim any controlling effects on pests/pathogens
<Our response> In Introduction, we added the information as: " Potassium phosphite salts (Agri-fos, Agrichem used here and other products with active ingredients of the same category) are products that have been registered in California for over 15 years as fungicides for the protection of oak and tanoaks trees from sudden oak death (SOD, causal organism Phytophthora ramorum) "
Round 2
Reviewer 2 Report
Thanks to the authors for taking my comments on board. I look forward to seeing the paper published.